# Peer review and preprint policies are unclear at most major journals

**Thomas Klebel**[1], **Stefan Reichmann**[2], **Jessica Polka**[3], **Gary McDowell**[4], **Naomi Penfold**[5], **Samantha Hindle**[6], **Tony Ross-Hellauer**[2]*

**1** Know-Center GmbH, Graz, Austria, **2** Institute of Interactive Systems and Data Science, Graz University of Technology, Graz, Austria, **3** ASAPbio, San Francisco, California, United States of America, **4** Lightoller LLC, Chicago, Illinois, United States of America, **5** eLife Sciences Publications Ltd, Cambridge, Cambridgeshire, United Kingdom, **6** Cold Spring Harbor Laboratory, bioRxiv, Cold Spring Harbor, New York, United States of America

* tross@know-center.at

**Data Availability Statement:** All files are available at http://doi.org/10.5281/zenodo.3743092.

**Funding:** The authors received no specific funding for this work.

## Abstract

Clear and findable publishing policies are important for authors to choose appropriate journals for publication. We investigated the clarity of policies of 171 major academic journals across disciplines regarding peer review and preprinting. 31.6% of journals surveyed do not provide information on the type of peer review they use. Information on whether preprints can be posted or not is unclear in 39.2% of journals. 58.5% of journals offer no clear information on whether reviewer identities are revealed to authors. Around 75% of journals have no clear policy on co-reviewing, citation of preprints, and publication of reviewer identities. Information regarding practices of open peer review is even more scarce, with <20% of journals providing clear information. Having found a lack of clear information, we conclude by examining the implications this has for researchers (especially early career) and the spread of open research practices.

## Introduction

Scholarly publishing, as the steward of the scientific record, has a great deal of power to steer researcher practices. Despite emergent trends towards greater openness and transparency in all areas of research [1, 2] publication practices of academic journals can remain something of a black box for authors and readers [3]. Processes of editorial handling and peer review are usually hidden behind curtains of confidentiality or anonymity. But worse, journal policies which should orient authors and readers as to the editorial standards employed by individual journals, including what the general type of peer review system is or whether preprinting manuscripts is allowed, have been suggested to be often unclear [4–6]. Unclear policies, for example regarding copyright or licensing, could expose researchers to unnecessary risk [5]. Lack of clarity of policies would also make it difficult for authors to find publishers with desirable practices, and even slow the appreciation among authors that different approaches are possible. Finally, opacity impedes our ability to track the prevalence of emerging policies, inhibiting understanding of how common and well-accepted those policies are.

This study aims to investigate the clarity of policies of major academic journals across academic disciplines regarding peer review and preprinting. With "preprint" we refer to either

**Competing interests:** Gary McDowell works at a for-profit that provides consulting services to organizations addressing issues concerning early career researchers. Samantha Hindle is Content Lead at bioRxiv, a preprint server for the biological sciences. Tony Ross-Hellauer is formerly Editor-in-Chief of the journal "Publications" (ISSN 2304-6775).

the submitted version (pre-print) or the accepted version (post-print) of an academic journal article (see the glossary from SHERPA/RoMEO: https://v2.sherpa.ac.uk/romeo/about.html).

Consider the case of a graduate student wanting to preprint their manuscript. The graduate student is concerned about publishing in a recognised journal, one they deem "high impact" so that they can make progress in their career. They may have to submit to several journals before their work is accepted for publication. Will preprinting preclude publication in any of these journals? The majority of researchers are disincentivised from preprinting if a journal does not accept preprinted submissions (59% of 392 respondents to ASAPbio survey, 2016, https://asapbio.org/survey). In reality, the majority of preprints posted to arXiv and bioRxiv end up being published in a range of journals [7, 8], and the graduate student can look up whether they can archive their paper, once accepted, using SHERPA/RoMEO The acceptance and adoption of preprints varies between disciplines: while established in several fields of physics [9, 10], computer science, and mathematics, adoption in the life sciences [e.g. 11–14], chemistry, medicine [15, 16], and the social sciences and humanities is lower, and this may affect how many journals explicitly encourage or allow preprinted submissions. Further, some journals may specify the type of preprint they allow: the specific server(s) it may be posted to, the licence used for the preprint, whether (and which) different versions may be posted, and what types of blog or media coverage of the preprint would constitute an unacceptable breach of any journal press embargo. Varied and vague policies make it harder for authors to understand what choices they have, and any constraints become more complicated with each additional journal considered. Furthermore, policies vary not only in their substance, but also in where they are communicated: sometimes they can be found under the instructions to authors, other times in many more obscure locations, and not unfrequently spread over several web pages. The path of least risk and resistance to the graduate student may simply be to not preprint at all.

The situation is more difficult if the researcher wants to select journals based on practices for which there are no databases, such as peer review practices (at least for journals that do not partner with Publons). If our graduate student prefers to submit to a journal that will anonymously publish the content of peer reviews (believing that these will be more constructive, well-prepared, and professional), they must assemble a list of candidate journals identified by word-of-mouth or by searching across multiple journal websites for policies that are often difficult to find. Various forms of innovation grouped under the umbrella term "open peer review" [17] result in a bewildering range of novel models for peer review. Especially for early career researchers, orienting themselves in this environment and understanding what is required of them can be a confusing process.

Finally, consider a graduate student deliberating whether or not to help their advisor with a peer review. They might want to know if a journal allows such *co-reviewing* and whether the review form enables them to be acknowledged when that review is submitted: in a recent survey, 82% of early-career researchers think it is unethical for PIs to submit peer review reports without naming all contributors to the report, and yet 70% of co-reviewers have contributed to peer review without any attribution [18]. The only way to find out if the journal process helps the graduate student's peer-review contributions to be recognised at present is to either contact the journal directly or to find someone with experience reviewing there.

TRANsparency in Scholarly Publishing and Open Science Evolution (TRANSPOSE) is a new initiative that addresses these issues. The TRANSPOSE initiative has created a database of journal policies for (1) open peer review, (2) co-reviewer involvement, and (3) preprinting (https://transpose-publishing.github.io). Here we undertake a closer investigation of a subset of journals to systematically taxonomize and analyse their peer review and preprinting policies as stated in journals' author guidelines. We surveyed 171 major academic journals, drawn

from the top-100 overall and top-20 per discipline of Google Scholar Metrics. The specific aims of the present study are to (1) systematically analyse the publicly available policies for pre-printing and peer review of a corpus of highly cited journals, (2) assess the clarity and explicitness of policies, and (3) provide evidence for best-practice recommendations. While desirable, policies are not located conveniently in a limited number of uniform documents in many cases. All journals in our sample make some form of author guidelines publicly available. However, availability does not make for understandability. The issue of policy clarity is particularly crucial for early career researchers or researchers new to a field. Senior researchers might have less trouble in navigating the journal landscape of a given field, likely having incorporated the fields' norms and practices, not least from prior experience with publishing in relevant journals. As indicated throughout the introduction, our analysis therefore takes the stance of a junior researcher trying to orient themselves within their field's publication landscape.

## Results

### Policy clarity

Within our sample, unclear policies are the norm, rather than the exception. We operationalize "clarity" pragmatically as whether a reasonably well-versed researcher would be able to *locate* and *understand* a given journal's regulations on peer review, preprints, and co-reviewing in a reasonable amount of time. Fig 1 displays all major aspects that were investigated, sorted by the proportion of clear policies within the sample. Overall, 54 out of 171 journals surveyed (31.6%) do not provide information on which type of peer review (double blind, single blind, not blinded, or other) is used. Information on whether preprints can be posted or not is similarly common, with 67 journals (39.2%) having no clear policy in this regard. There is no clear information on whether reviewer identities are revealed privately to the authors for 100 out of 171 journals (58.5%). Three quarters of journals in our sample have no clear policy with respect to whether co-reviewing is allowed, whether preprints can be cited or if reviewer identities are published. All other aspects (listed in Fig 1) are even more unclear, with 80% to 90% of journals giving no clear information on their website.

Regarding policy clarity, there is substantial variation between disciplines and publishers. This gives rise to many relevant questions: In what ways are policies related to each other? Do journals that allow co-reviewing also allow preprints? Is there a gradient between journals that encourage open research, and others that don't? Or are there certain groups of journals, open in one area, reluctant in another and maybe unclear in a third? To answer these questions, we employ Multiple Correspondence Analysis (MCA).

Results indicate that the different aspects of open research policies go hand in hand (Fig 2A, S1 Table). Journals with clear policies on posting preprints tend to also give clear information on whether coreviewing is accepted, which type of peer review is used, and whether reviewer identities are revealed to the authors. On the other hand, journals with unclear policies in one area more often than not have unclear policies in the other areas. Dimension 1 (horizontal) in Fig 2A represents this gradient from journals with above average clear policies to journals whose policies are less clear than the average. This first dimension accounts for 72.2% of total variance, while the second dimension only accounts for 4.1% of total variance. The second dimension is thus of relatively small importance and should only be interpreted with caution [see also 19]. It mainly represents journals that have clear policies on co-reviewing and unclear policies on posting preprints on the bottom, with the complementary journals on top.

Fig 2B displays differences between disciplines and publishers projected onto the first dimension. Overall, the gradient between journals with clear and unclear policies, respectively, is aligned along the distinction between journals from Science, Technology, Engineering and

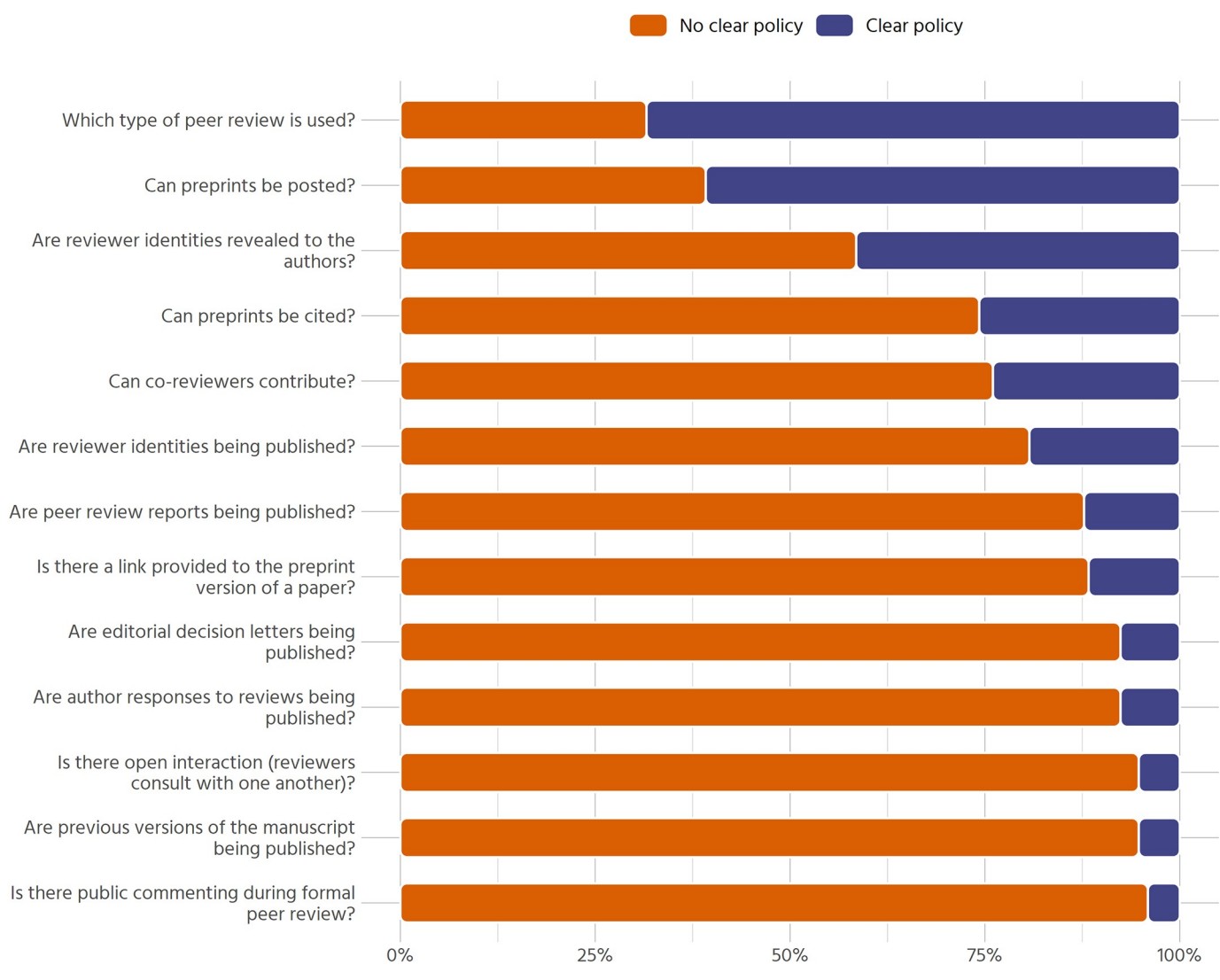

**Fig 1. Overall clarity of policies (n = 171).**

Mathematics (STEM) and Medicine, which are in most cases clearer than the average journal, and journals from the Social Sciences and Humanities (SSH), which are less clear than the average journal. Journals from the life sciences and earth sciences are well above average regarding clarity of policies, with journals from physics & mathematics, chemical & materials sciences and health & medical sciences being slightly above average. Journals from engineering & computer science are slightly below average, followed by journals from the social sciences, and humanities, literature & arts. From the junior researcher's point of view, journals from business, economics & management have the least clear policies of our sample. The publishers of the journals sampled broadly reflect these disciplinary differences. Journals from Springer Nature and the Royal Society of Chemistry are well above average with regard to policy clarity. While the American Chemical Society is close to the average of journals sampled with respect to policy clarity, Elsevier, IEEE, and those publishers in the "other" category are below average with regard to clarity of policies. The journals published by SAGE and Wiley do not adhere to this overall trend. Although all journals by SAGE in our sample belong to the Social Sciences

A

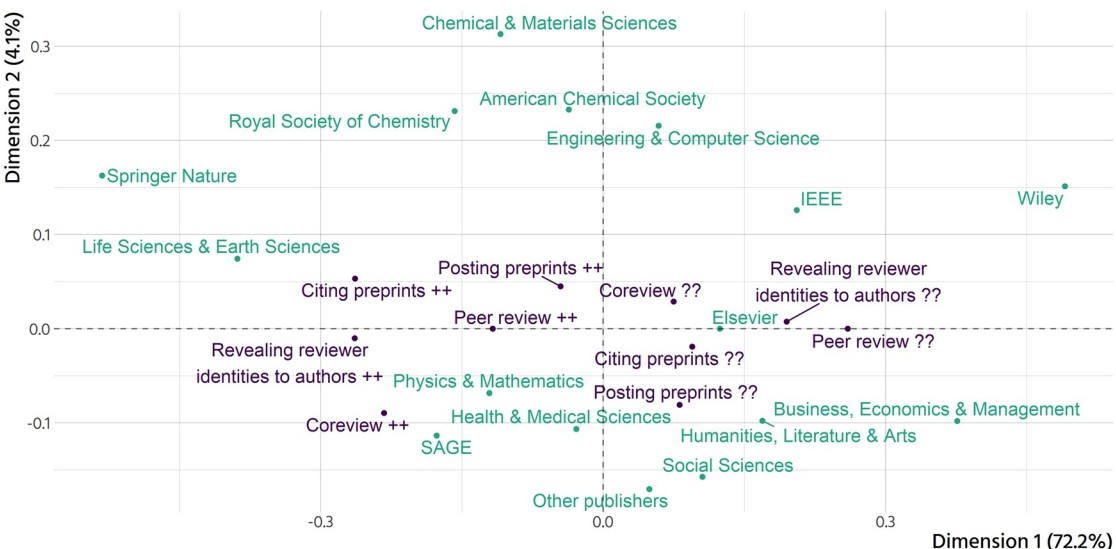

B

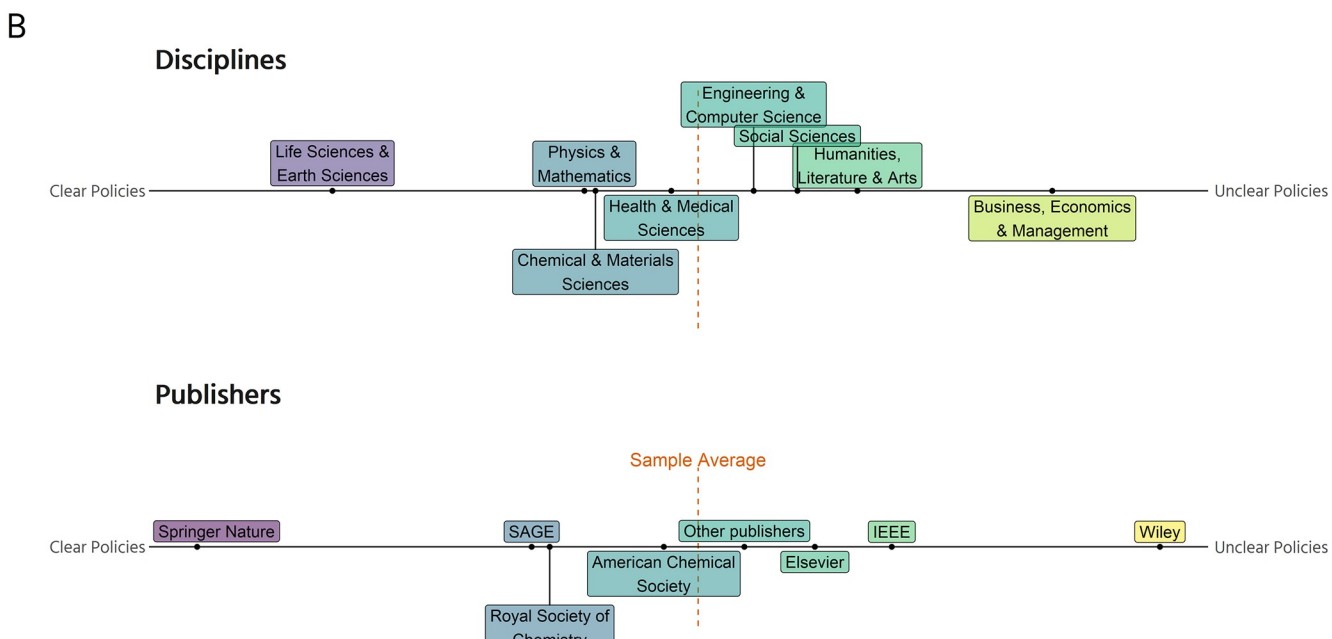

**Fig 2. The landscape of open research policies.** (A) Graphical display of a Multiple Correspondence Analysis. The contributing variables are the basis for the model and determine the layout of the space. "++" means that there is a clear policy, "??" that there is no clear policy. Disciplines and publishers were added as supplementary (passive) variables and have no impact on the space. Dimension 1 (horizontal) explains 72.2% of the variance, Dimension 2 explains 4.1% of the variance in the contributing variables. (B) The supplementary variables from (A) projected onto the horizontal axis. Journals from disciplines and

publishers with policies that are clearer than the average journal in our sample are located on the left, journals with less clear policies than the average on the right. Elsevier includes journals published by Cell Press, Wiley journals published by Wiley-VCH.

and Humanities where policies are comparatively unclear, they are much clearer than the average journal. On the other hand, journals published by Wiley which in our sample come from a broad range of disciplines, are particularly unclear in their policies compared to all other journals sampled.

## Peer review

Availability of information on the type of peer review used by a journal is mixed (Fig 3A). For those journals with clear information, the most common peer review policy is single blind peer review (29.8%), followed closely by double blind peer review (26.9%). Some journals offer the option for authors to choose whether to use single or double blind peer review–for example, the Nature journals have a single-blind process as default but allow authors to choose to be double-blind if preferred. These cases have been coded as "Other", accounting for the majority

A

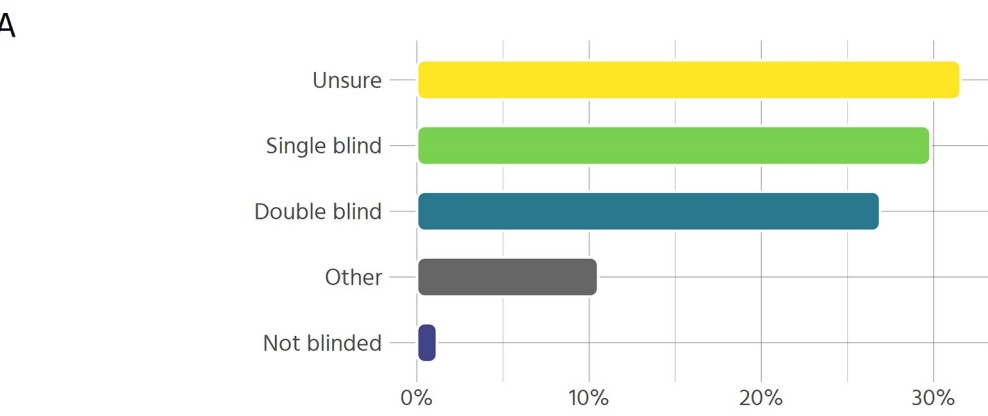

B

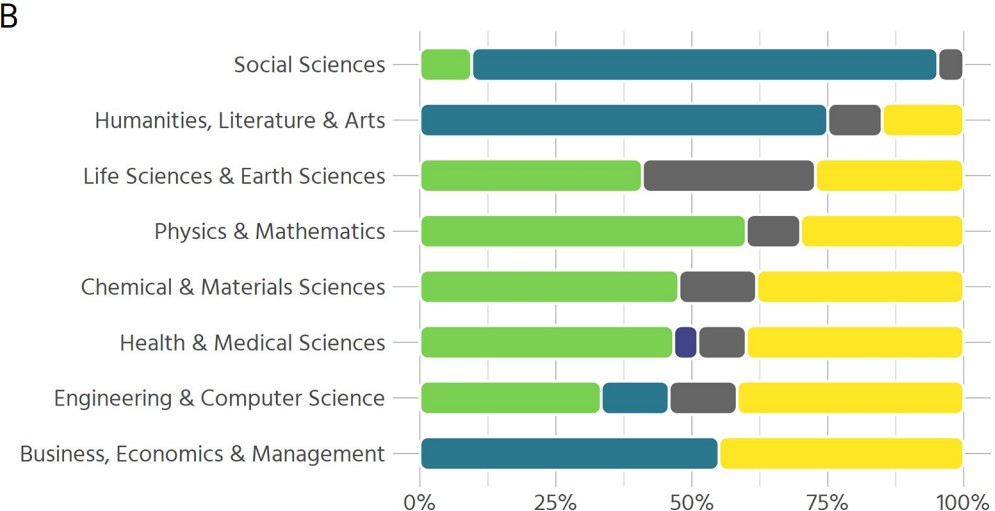

**Fig 3. Type of peer review employed by journals.** (A) Type of peer review used overall (n = 171) (B) Type of peer review used by disciplines (n = 193).

of titles belonging to this category. 1% of journals ("The BMJ" and "The Cochrane Database of Systematic Reviews") do not anonymize authors or reviewers during the review process.

However, there are major differences between disciplines (Fig 3B). In the social sciences, humanities, and business, double blind peer review is the norm, while the natural sciences rely more heavily on single blind peer review. Among all disciplines, business, economics & management display the highest proportion of unclear policies, with social science and humanities being very clear and the remaining disciplines somewhere in between.

## Open peer review

Information on open peer review (OPR) is similarly scarce (Fig 4A) across the sample. The survey included questions on select dimensions of OPR, e.g. whether a journal publishes peer review reports, editorial decision letters or previous versions of the manuscript, whether it offers public commenting during the peer review process, and similar questions. More than 50% of journals surveyed do not provide any information on these aspects of OPR. No journal in our sample allows public commenting during formal peer review. Other forms of openness are similarly rare. With the exception that some journals state that they may reveal reviewer identities to authors, information on the other aspects is either not specified or OPR is not practiced by more than 95% of journals.

As revealing reviewer identities privately to authors is the only aspect of OPR that is explicitly allowed by a substantive number of journals (23.4%), we examine it separately for each discipline (Fig 4B). Whereas the social sciences, humanities and business journals' policies do not mention revealing reviewer identities to authors, this is not unusual in the natural sciences, at least on an optional basis (many journals offer referees the opportunity to sign their reviews).

## Co-review

Information on co-review policies is not uniformly available: 87 out of 171 journals (50.9%) have an explicit co-review policy. There are notable disciplinary differences (Fig 5). In the life and earth sciences, health & medical sciences as well as physics & mathematics more than a quarter of journals explicitly permit contributions from co-reviewers, whereas in the humanities, chemical & materials sciences, and in business, economics & management around 10% do.

To obtain a more nuanced view of the policies, we analysed their content via text mining. Table 1 displays the most frequent terms of the distinct policies (n = 35), sorted by the proportion of policies that contain a given term. The terms were stemmed prior to counting them so occurrences of similar meaning would count towards the same term (like confidential/confidentiality counting towards confidenti). The most prominent themes that emerge are:

- Individuals with varying stakes regarding peer review: editor, colleague, collaborator, student, author, peer.

- Confidentiality as a central principle.

- Important elements of scholarly publishing: manuscript, journal, review.

- Verbal forms pertaining to relationships between the individuals: inform, involve, consult, discuss, disclose, share.

Journals stress the importance of "maintaining confidentiality" through "not shar[ing]" or disclosing information, neither to "junior researchers" and "laboratory colleagues" nor to "graduate students" (see also S1 Fig). Even if the policies do not explicitly forbid or allow the

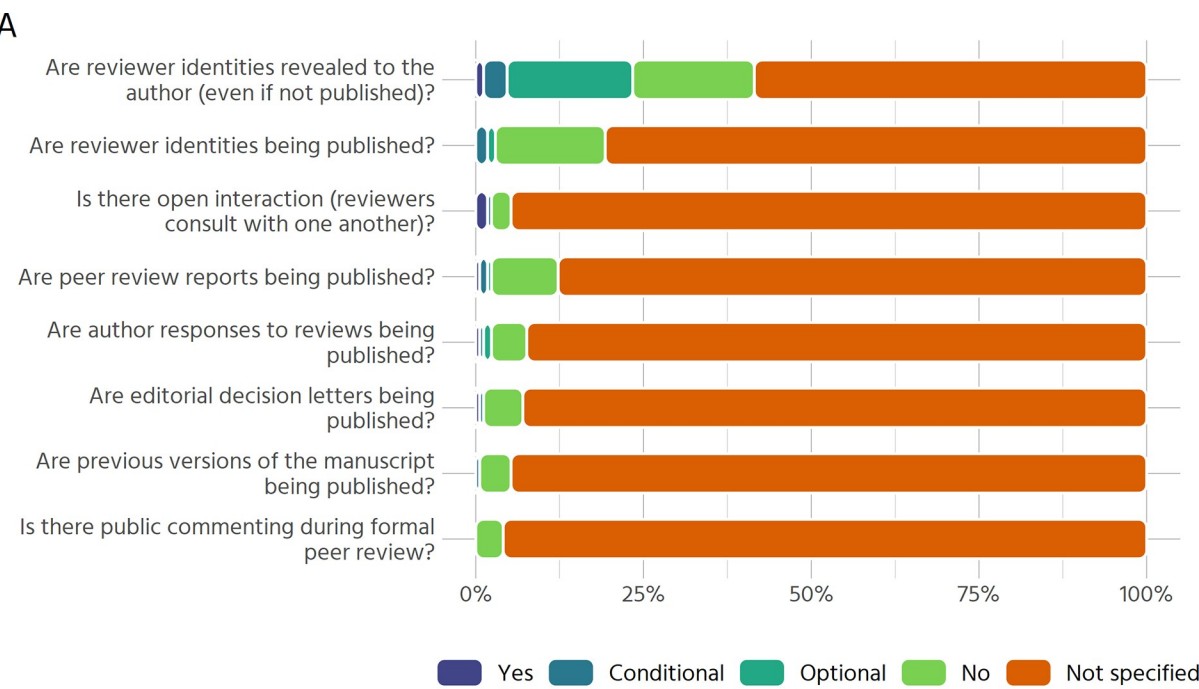

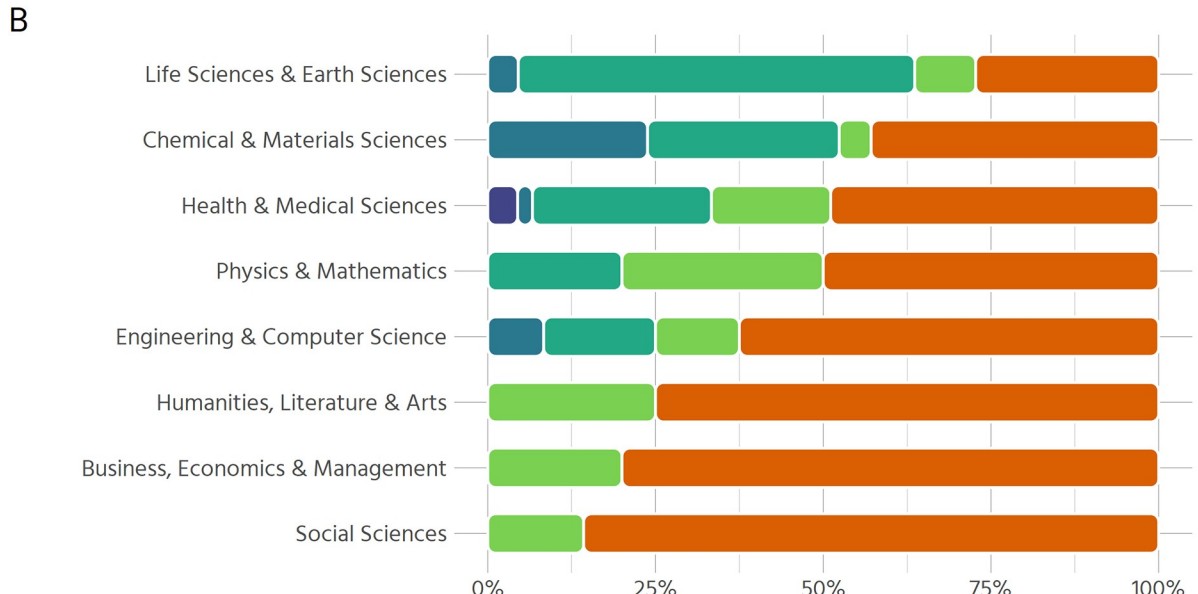

**Fig 4. Aspects of open peer review.** (A) Aspects of open peer review across all journals in the sample (n = 171). Categories: Yes, Conditional (i.e., is true if other conditions apply), Optional (i.e., either author or reviewer can choose but not mandatory), No, Not Specified (i.e., information not found online) (B) Results on whether reviewer identities are revealed to the authors, even if they are not published (n = 193).

involvement of other researchers, in many cases they mandate the reviewer to first obtain permission from the editor in case they want to involve someone else in their review. The editor's prominent role can also be observed by the terms' frequent appearance in the policies: almost three quarters of all policies mention the term "editor". In the majority of cases, policies state

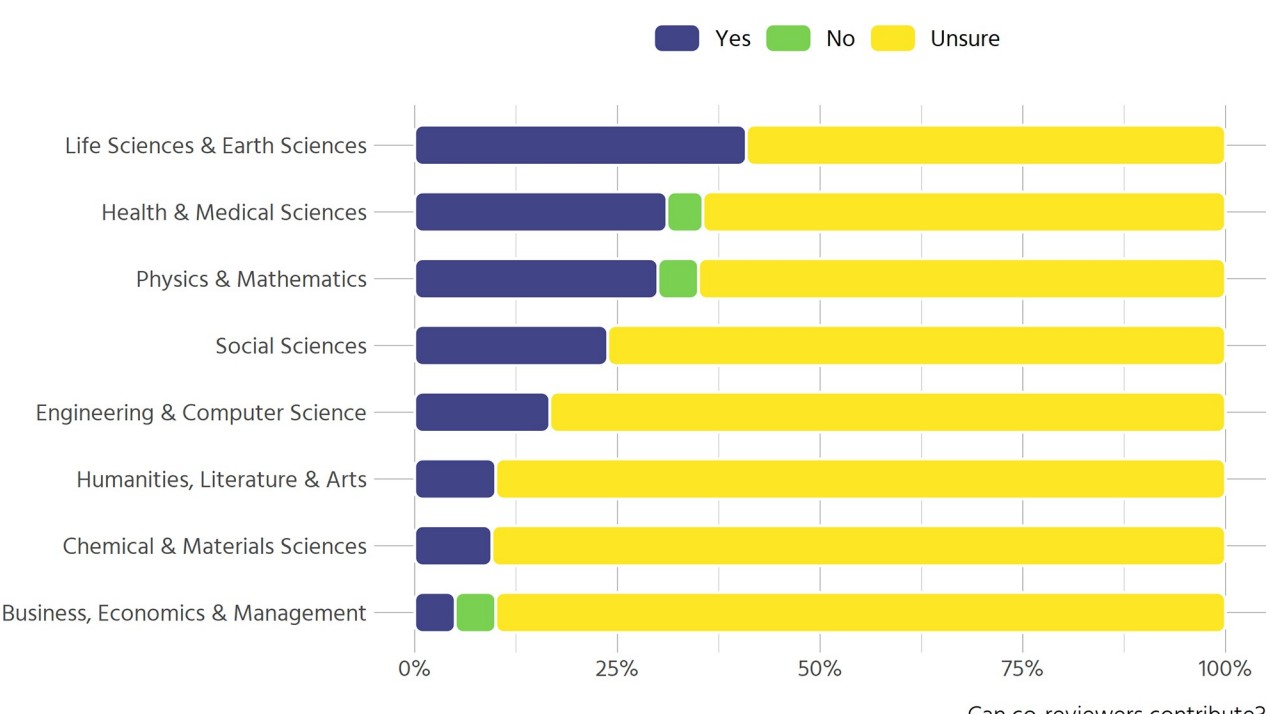

**Fig 5. Clarity of co-review policies (n = 171).**

that one must "obtain permission from the journal editor" to show the manuscript to others or that co-reviewing is not permitted "unless previously agreed with the editor".

## Preprints

Policies for posting or citing preprints are more common within our sample compared to open peer review or co-review policies. 120 out of 171 journals (70.2%) state that they allow some form of preprints. Most of them (39.2% of the total sample) allow preprints before peer review while 22.8% do not have a preprint policy.

Similar to our results on peer review, preprint policies vary considerably between disciplines (Fig 6A). While in the life sciences & earth sciences 91% of all journals allow preprints in some way, in the humanities 45% do. The natural sciences in general tend towards allowing preprints only on first submission (before peer review). Journals from the social sciences, the humanities and from business, economics and management generally either have no preprint policy at all or are more diverse in regard to preprint version, also allowing preprints after peer review.

A complementary aspect of the acceptance of preprints is whether they can be cited. The majority of journals (57.3%) do not specify whether this is possible. Unclear policies on how to cite preprints (e.g. in the references or only as footnotes in the text) are also quite common (15.2%). Where citations of preprints are allowed, this is possible in the references for 78% of journals, with some journals restricting citations of preprints to the text (14%).

Preprint policies with respect to citations again vary greatly between disciplines (Fig 6B). Policies permitting citation of preprints are more common in the natural sciences, with 55% of all journals in the life and earth sciences allowing citations to preprints either in the text or in the reference list. In contrast, the social sciences and humanities largely have unclear policies or no policies at all regarding whether preprints can be cited or not.

**Table 1. Propensity of terms in co-review policies.**

| Term | Variants | Term frequency | Proportion of policies that contain term |
|---|---|---:|---:|
| review | review; reviewers; reviewer | 100 | 93% |
| manuscript | manuscript; manuscripts | 43 | 75% |
| editor | editor; editors | 33 | 73% |
| confidenti | confidential; confidentiality | 26 | 63% |
| not | not | 24 | 60% |
| inform | information; inform; informed | 19 | 51% |
| colleagu | colleague; colleagues | 18 | 49% |
| student | students; student | 14 | 34% |
| discuss | discuss; discussed; discussion | 12 | 32% |
| involv | involved; involve; involving | 12 | 32% |
| consult | consult; consulted; consulting | 12 | 32% |
| permiss | permission | 11 | 31% |
| disclos | disclosed; disclose | 12 | 29% |
| author | authors; author; authorization | 11 | 29% |
| peer | peer | 10 | 29% |
| journal | journal | 10 | 28% |
| share | share; shared; sharing | 9 | 25% |
| collabor | collaborate; collaborators; collaborating | 10 | 24% |
| advic | advice | 8 | 23% |
| ident | identities; identity | 8 | 23% |

Terms in the column "Terms" were stemmed using the function 'wordStem'from the SnowballC R package. "Variants" displays the three most common variants for a given term as they appear in our data. Sometimes only one variant (e.g. peer, journal) was present in the data. Further context is provided in S2 Table which lists one sampled sentence for each term variant.

Besides investigating policies on posting and citing preprints, we surveyed other aspects of preprint policies as well: whether there is information on which licenses are permitted for the preprint, or if there is scoop protection, e.g. if a preprint will still be considered for publication even if a competing work is published in another journal after the date of preprinting. Further aspects were whether a published paper includes a link to the preprint version, what type of media coverage of the preprint is permitted and if there is a policy on community review for preprints. Overall, guidance on these issues is rarely provided: 72.5% of journals provide no information on permitted media coverage and 88.3% of journals provide no information on whether the publication will include a link to the preprint. 94.7% of journals provide no guidance on which license is permitted for the preprint, 98.2% give no information on scoop protection, and 98.2% of journals give no indication whether public comments on preprints will have any effect on manuscript acceptance.

## Discussion

### Clarity of journal policies

Our results suggest that policies regarding various aspects of scholarly publishing are very often unclear. Even the most basic kind of information–which type of peer review a journal uses– could not be found on the website in more than 30% of journals. Information on all other aspects we investigated is even harder to find. This is problematic, since it hinders the uptake of open research practices on several fronts. Authors might be reluctant to post or cite preprints if they cannot be sure how this will impact their submission. Reviewers might be

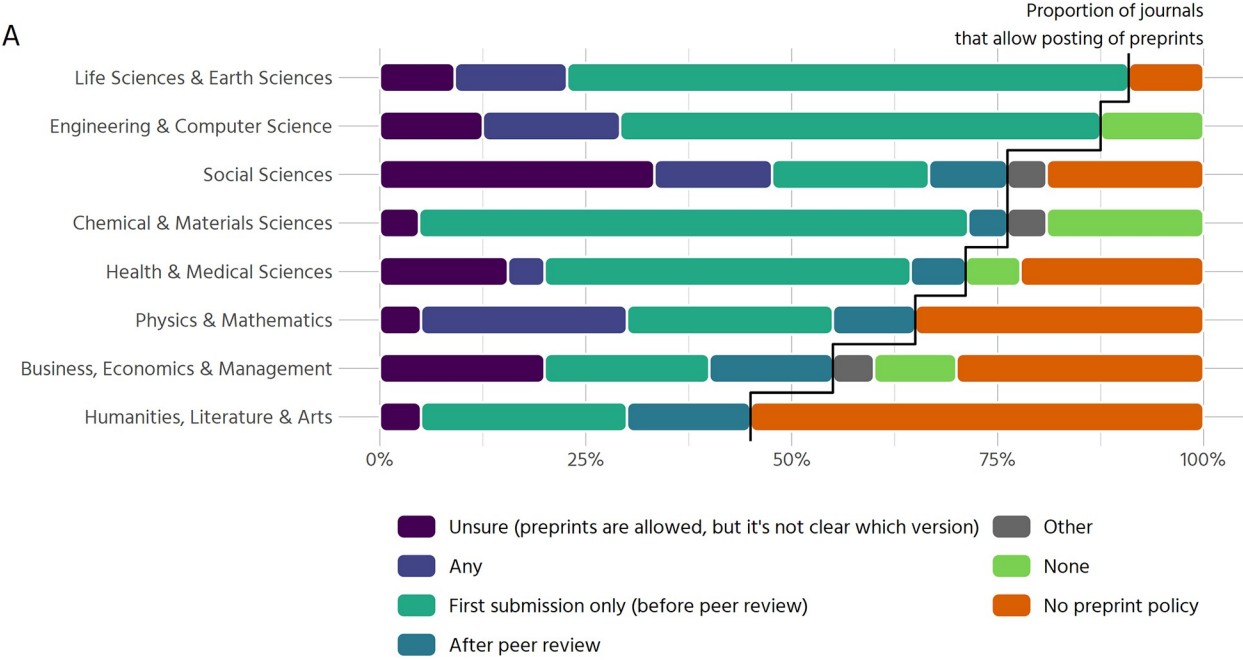

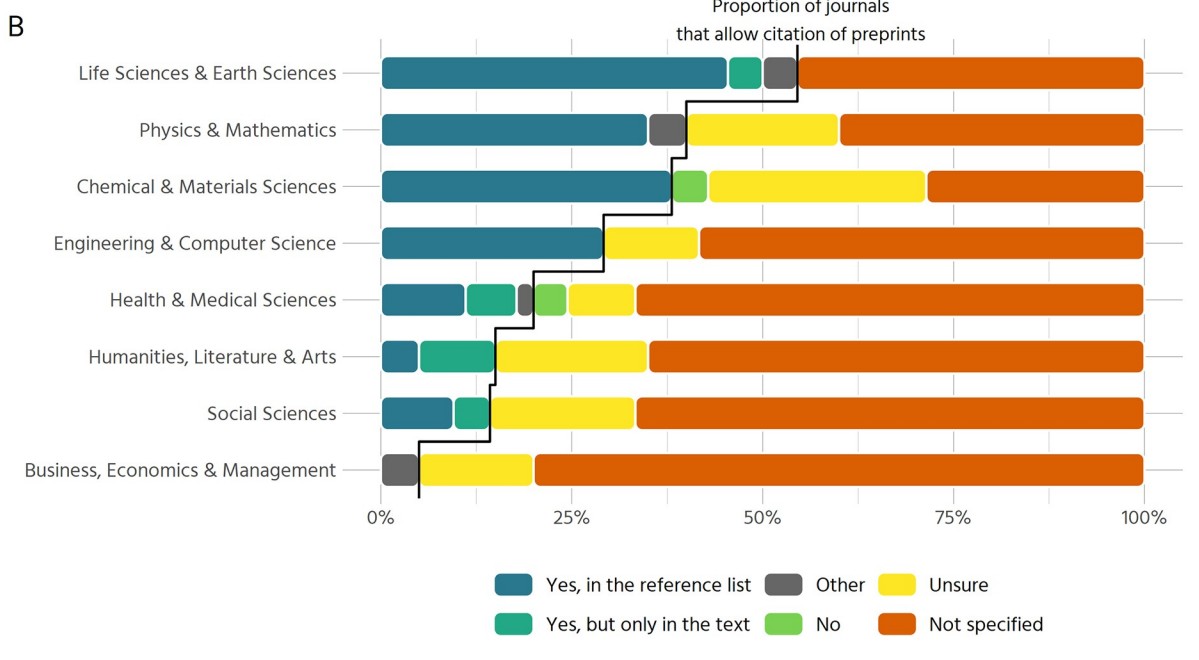

**Fig 6. Posting and citing of preprints.** (A) Results on whether a preprint can be posted, and which version is allowed (n = 193). (B) Results on whether preprints can be cited (n = 193).

disinclined to sign their reviews or involve junior colleagues in writing the review if they do not know how editors will handle these cases.

We found that there is a gradient between journals that have clear policies on the different aspects of open research practices and other journals with unclear policies. This gradient is roughly structured along the distinction between SSH and STEM disciplines. Since open research practices are as yet less common in the SSH, it should come as no surprise that journals have no or unclear policies. The other side of the gradient is marked by disciplines from the natural sciences where, generally speaking, open research practices are more common [9–12, 14–16].

An alternative explanation to the lack of clear policies might also be that a given practice (e.g. double blind peer review) is so common in certain disciplines that specific policies are not put in place or not communicated transparently. This is especially relevant for early career researchers which likely are less aware of a field's norms and practices. One of our findings helps to illustrate this point. Recall Fig 4B, where we investigated whether reviewer identities are revealed to authors, even if they are not made public. The high proportion of journals within SSH that are categorised as "Not specified" might be surprising, given that most of them conduct double blind peer review. One could thus infer that reviewer identities are not revealed to the authors. However, it might be countered that this reveals the root problem: there is no clear policy. Reviewers might sign their review or not; what the authors receive is at the editor's discretion. Peer review and co-review.

We found that 31.6% of journals in our sample don't offer clear information on which type of peer review they employ. This is in line with Utrobičić et al. [20] who studied editorial structures and peer review policies in Croatian journals indexed in Web of Science, finding a lack of transparency of publicly available information for authors on peer review processes. There are ongoing debates e.g. in medical journals how this situation might be amended [e.g. 4, 21]. Increasing availability of information regarding the editorial procedure might be beneficial for journals themselves, since disclosure of information about the editorial and peer-review process correlates with authors' perceptions of a high-quality peer review process and the journal rejecting hoax papers [22].

The highly influential role of editors in what practices are acceptable or prohibited and how certain policies might be implemented has been investigated with regard to peer review [23]. This can be extended to the issue of co-reviewing. 50.9% of journals in our sample have an explicit co-review policy. Analysing the respective policies revealed that many of them reference confidentiality as a core principle. If a manuscript is to be shown to or discussed with another researcher, reviewers have to ask the editor for permission in the majority of cases. This is problematic, since co-reviewing and ghostwriting is very common among early career researchers, and in practice permission is not asked for from the editor, but the manuscript is shared anyway [18]. Early career researchers will likely hesitate to contact the journal's editor if their superior asks them to help with or write a review, and in turn the invited reviewer will, upon the submission of the review, consider omitting the participation of the co-reviewers as the lesser sin compared to not having asked permission to do so–or simply may not consider naming of co-reviewers as necessary, in the absence of clear journal policies surrounding co-reviewing. In addition, the contribution of early career researcher co-reviewers might be prohibited by informal editorial policy or it might go unnoticed, since acknowledging the efforts made by multiple reviewers is very rare in general.

## Preprints

Researchers generally feel they must publish in community-recognised journals for career progression and evidence of productivity. As a consequence, whether a journal regards preprints

as prior publication or not is an important policy factor, as posting a preprint of a manuscript might effectively forestall publication in a journal. Additional considerations where authors may expect clarity include preprint licensing, which version can be uploaded to which server (s), and whether preprints can be cited (and if so, how). All of these considerations matter to the individual author as well as to the use of preprints in a discipline in general. We found that 39.2% of journals sampled do not offer clear information on whether preprints can be posted online, and if yes, whether before and/or after submission to the journal. This percentage is substantially higher than Teixeira da Silva & Dobranszki [13] found using data from SHERPA/ RoMEO. They report 80.3% of publishers in their sample permitting self-archiving of manuscripts. The difference is likely due to differing perspectives on preprints: SHERPA/RoMEO only holds information on which version of a paper (pre-print or post-print, i.e. the manuscript before or after peer-review) can be archived. Whether a manuscript that has been posted to a preprint server prior to submission will still be considered for publication by any given journal is not recorded by SHERPA/RoMEO but has been examined by our study and is reported in the TRANSPOSE database (https://transpose-publishing.github.io). While permitting posting of preprints is very common in our sample, the majority of journals (57.3%) do not specify whether a journal permits *citing* preprints.

The content of preprint policies varies by discipline. For example, in the humanities only 45% of journals explicitly allow authors to post preprint versions of their manuscript, while in the life and earth sciences 91% do. Our results in this regard support previous work on disciplinary cultures and differential propensity to accept preprints [e.g. 24]. In the social sciences, publication patterns and citation cycles differ markedly from those in the natural sciences, e.g. citation cycles are generally much longer [25], reducing the efficacy of preprinting. Furthermore, the social sciences and humanities operate on vastly different conceptions of originality [26], placing different strains on publication processes.

In summary, we find that policies regarding various aspects of scholarly publishing are very often unclear or missing. This is not to say that policies should be an iron cage, with no flexibility for editorial decisions. Professional judgement is an important part of performing the tasks of an editor. However, uncertainty for authors and reviewers alike is unconstructive. If there is no guidance on whether certain practices are encouraged or prohibited, submitting and reviewing for journals becomes a minefield that is not easily navigated. This might further hinder scholarly participation from early career researchers who are less accustomed and aware of certain norms in their field.

## Data and methods

We used the Google Scholar Metrics service (GSM) to compile a list of the top 100 publications (journals) ordered according to their five-year h-index metric as of 13th October 2018 (query: https://scholar.google.co.uk/citations?view_op=top_venues&hl=en). The five-year h-index "is the largest number h such that h articles published in [the last 5 complete years] have at least h citations each" [27]. In addition, we took the top 20 results from each of the 7 broad subcategories offered by GSM: Business, Economics & Management; Chemical & Material Sciences; Engineering & Computer Science; Health & Medical Sciences; Humanities, Literature & Arts; Life Sciences & Earth Sciences; Physics & Mathematics; Social Sciences. Results were returned on 13th October 2018 (although the GSM about page notes these results are based on "our index as it was in July 2018"). These lists were copied over to a spreadsheet where the journal titles were compiled and de-duplicated, with information retained about their relative position in one or more of the top 100 and 7 sub-categories. The full list is available at https://zenodo.org/record/3959715.

We acknowledge several limitations of this approach. Firstly, GSM does not enable browsing by subject area for non-English-language titles. This naturally means that our lists do not properly represent non-English language titles. Moreover, by focusing on "high-impact" titles, we can assume our sample is biased towards titles that are better resourced financially, which can be assumed to have more developed policies in place than their less well-resourced counterparts. Hence, this landscape scan cannot represent the totality of the journal landscape. In addition, this is based on non-open data: the criteria for inclusion and exclusion in the Google Scholar index are opaque and non-reproducible [28]. This study, however, does not aim at a complete picture of all journals across all domains, regions and languages–rather to scope the policies of a limited number based on their perceived prominence to global scholarly communities, and with a corpus that is manageable for qualitative investigation and classification. The h-index has been subject to critique regarding its use as a measure of scientific impact [29]. Here, however, we are clear that it is used only as a proxy for visibility within scientific communities. A further difficulty with this approach is that taking only the top 20 journals in each category further impacts the representativeness of this sample. Levels of citations vary widely not only between broad categories of research, but also within specific disciplines and subdisciplines [30], and the number of journals sampled does not scale with the total number of journals or researchers in those areas. Again, we here acknowledge this limitation as an artefact of the pragmatic need to compile a corpus small enough to allow qualitative interrogation but large enough to include at least some data on differences across broad categories of research. We of course encourage further replications of this analysis at subdiscipline level.

## Data collection

De-duplication returned a list of 171 journals. Each title was then assigned to two assessors who applied a standardised data-collection instrument and protocol to determine what information is publicly available online regarding peer review and preprint policies at each journal. The first round of data-collection took place between 2018-11-21 and 2019-02-15 and the second round between 2019-04-11 and 2019-04-24. In a third round between 2019-04-24 and 2019-04-28 data from the two assessors was cross-checked, resolving any discrepancies. The data-collection instrument is available at https://zenodo.org/record/3959715. The aim was to mirror the experience of a researcher who might wish to find this information online. Search began from the journal website, and internal links followed from there. No secondary sources were used (e.g., assessor's prior knowledge; external databases; contact with journal editorial staff). An alternative strategy was to use web keyword search (via Google) using, for example "[journal name] AND 'peer review' OR 'pre-print' OR 'preprint' OR 'working paper'", or, in the case of co-reviewing policies, "[journal name] AND 'confidentiality'". The second assessor checked the first assessor's answers and revised or challenged based on their own interpretation of the information found online. Disputes were then adjudicated by two authors (JP & TRH) who reviewed the second-round edits in a third and final round. Note here that we do not claim that our dataset collects all possible information which could have been found online for these journal policies. Information can be spread widely over a confusing number of journal- and/or publisher-level pages. Hence, there is the possibility that some information was not captured despite two rounds of review.

After the third round of review, the collected data were imported to R and cleaned for further analysis. This involved unifying categories for plotting and merging with data from GSM on disciplinary area. The approach taken to create the sample of journals led to a few journals having no subdiscipline: some journals like "Gut" were within the top 100 journals, but not within any of the subdisciplines. This is because the h5-index varies widely between subdisciplines. Fig 7A shows the top-20 journals of each discipline.

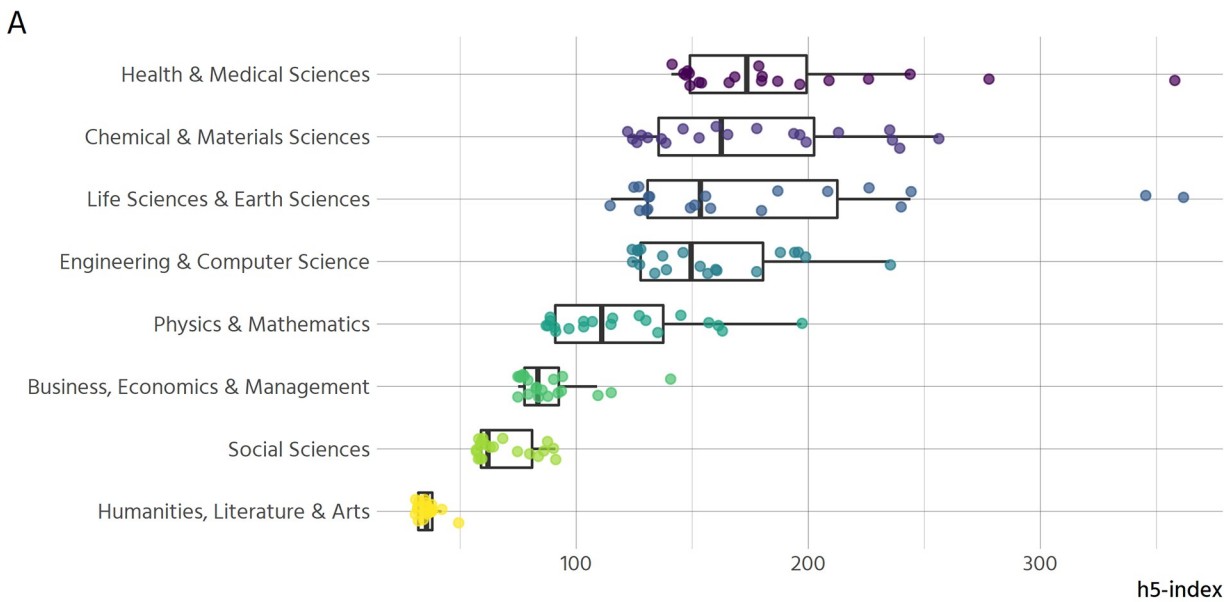

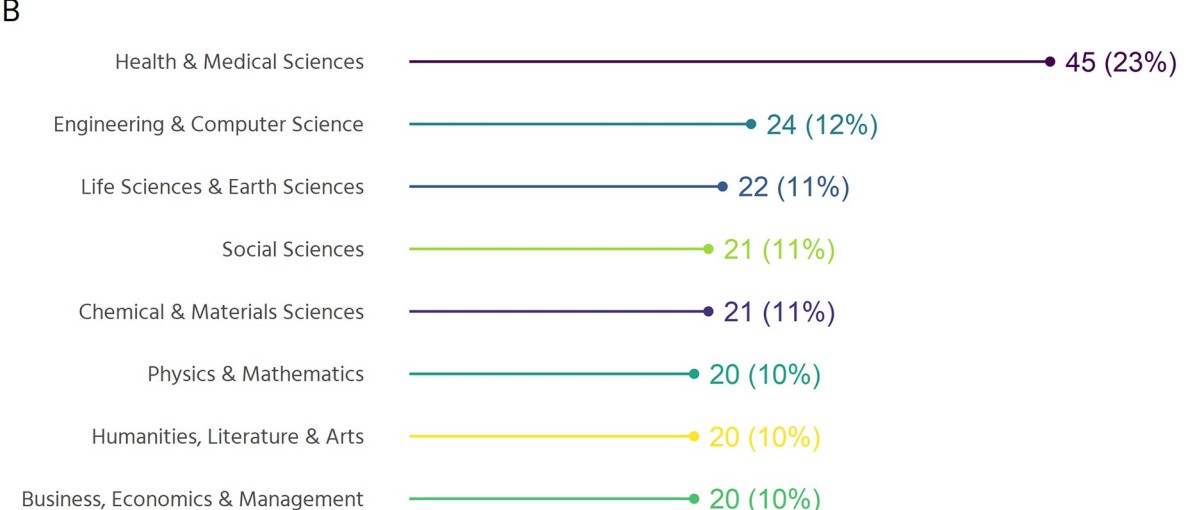

**Fig 7. Sample characteristics.** (A) The distribution of journal's h5-indices across disciplines (n = 171). (B) Number and proportion of journals sampled by discipline (n = 193).

The missing categorisations were added in a second step, to facilitate analysis of all journals that distinguishes by discipline. To this end, we scraped all disciplines and subdisciplines from GSM on 18th of June 2019 and matched those to our data.

As stated, the criteria for inclusion into the Google Scholar rankings are opaque and non-reproducible. For example, it is possible for a journal to be included in different disciplines, e.g. "Physics & Mathematics" along with "Engineering & Computer Science". It is however also possible for a journal to be included in a subdiscipline, and not in the parent discipline, despite having a higher h-index than all journals listed in the parent discipline (e.g. as of 2019-12-20, the "Journal of Cleaner Production" is listed in the social sciences under "sustainable

development" (https://scholar.google.at/citations?view_op=top_venues&hl=en&vq=soc_
sustainabledevelopment). But it is not listed under the parent category (https://scholar.google.
at/citations?view_op=top_venues&hl=en&vq=soc)).

The nature of our selection means that 22 out of 171 journals are assigned to two disci-
plines. All results that distinguish between disciplines are therefore based on 193 cases. The
inclusion criteria further mean that disciplines are not represented equally within the sample.
Since about one quarter of the top 100 journals belong to the health and medical sciences, the
sample is slightly skewed in that direction (Fig 7B).

Regarding practices of open access, only 8 of 171 journals are listed in the Directory of
Open Access Journals (DOAJ) and can thus be considered fully open access.

## Methods

Data analysis was done in R [31], with the aid of many packages from the tidyverse [32]. The
analysis of the policies generally follows two directions: first, whether clear policies can be
found, and second, what their content is.

To investigate clarity on policy, we selectively recoded variables with regard to whether cer-
tain policies were clear or not, thus omitting the subtle differences within the policies (i.e. "which
version of a preprint can be cited" was simplified for whether the policy was clear (references
allowed in text, reference list or not allowed) versus unclear (unsure about policy, no policy and
other)). "Clarity" of author guidelines therefore has been operationalized pragmatically as
whether a reasonably well-versed researcher would be able to *locate* and *understand* a given jour-
nal's regulations on peer review, preprints, and co-reviewing in a reasonable amount of time. It
should be noted that this represents an analytic categorization which is not necessarily reflected
in the conceptualizations employed/relevance ascribed by journals. However, we expect any
assessor with reasonable practical knowledge of academic publishing to be able to reproduce the
data collection procedure based on the assessment framework described in the section "Data
Collection". It should be noted, though, that conducting the data collection procedure again will
lead to partly different results, since the policies under scrutiny are subject to change.

After recoding for clarity, we analysed the variables via Multiple Correspondence Analysis
[33], which lets us explore the different policies jointly [34] and thus paint a landscape of open
research policies among journals. It should be noted that this procedure is strictly exploratory.
We are exploring possible associations between the policies, not testing any hypotheses.

We included five active categories in our model. All were recoded in terms of whether there
was a clear policy on:

- Type of peer review (double blind, single blind, not blinded, or other)

- Co-reviewing

- Revealing reviewer identities to authors

- Posting preprints

- Citing preprints

The geometric layout of the space displayed in Fig 2A is determined by these five active cat-
egories. Interpretation of the points displayed is achieved by projecting them onto the axes.
Furthermore, only statements regarding the sample average are possible. If a given journal is
far away from zero towards the left (right) part of the figure, it indicates that this journal's poli-
cies are more or less clear than the rest of the sample, but not that the journal's policies are
clear or unclear in absolute terms. To further illuminate some of the results, the disciplinary

areas and the five most common publishers were added as passive categories. They have no influence on the geometric layout but allow us to draw conclusions on which policies are more prevalent in one area or another.

To investigate the policies' contents, the main analytical approach was to create displays of cross tabulations with ggplot2 [35]. When reporting percentages from these cross tabulations, we report percentages with one decimal for the full sample (171 or 193 journals (e.g. 23.3%)). When reporting disciplinary differences (n = 20–45), we report percentages without decimals (e.g. 23%).

Co-review policies were further analysed via text mining. Due to the prevalence of publisher-level policies for many journals in the sample, there are 35 distinct policies on co-review in our dataset, compared to 87 policies in total. During data collection, investigators manually copied relevant parts of the policies to our dataset. This inhibited detecting duplicates, since for the same policy different parts might have been copied or abbreviated. To identify duplicates, TK compared the policies with the distance metric "Jaccard" and then manually went through the most similar ones. Selection for deleting duplicates was done by keeping the version with more text to retain as much information as possible. Since the policies are generally rather short in length, the analysis is somewhat limited with respect to insights we can gain from automated procedures. To extract meaningful information we first removed common words from the English language (via the list of stop-words from the tidytext package [36], except for the word "not", which is relevant since some policies state that it is *not* appropriate to share information with students or colleagues). The resulting list contains 886 words in total. For a simple overview, the words were stemmed to reduce similar but not identical versions of certain words (like editor/editors).

We used the package visdat [37] to explore the data at the beginning of analysis, and used ggrepel [38] to design comprehensible figures with labels. All data and code, including a reproducible version of the results section, is available at https://zenodo.org/record/3959715.

## Supporting information

**S1 Fig. Directed bigram graph of co-reviewing policies.** Displayed are bigrams for all terms in the co-reviewing policies, after removal of stop-words (the word "not" was not removed, see the "Methods" section). When creating bigrams, the text is split into pairs of words (for example the sentence "All humans are equal" becomes "All humans", "humans are", "are equal"). The most prominent bigrams were "peer -> review" and "review -> process". To look at the strength of other associations, the term "review" was removed from the figure. The most frequent associations in the figure are depicted by bold arrows.
(TIF)

**S1 Table. Numerical output from multiple correspondence analysis.**
(PDF)

**S2 Table. Sample phrases for prominent terms in co-review policies.**
(PDF)

## Acknowledgments

We thank the 2018 Scholarly Communication Institute (Chapel Hill, NC) for supporting the early phases of this project.

## Author Contributions

**Conceptualization:** Jessica Polka, Tony Ross-Hellauer.

**Data curation:** Thomas Klebel, Stefan Reichmann, Jessica Polka, Gary McDowell.

**Formal analysis:** Thomas Klebel.

**Funding acquisition:** Tony Ross-Hellauer.

**Investigation:** Thomas Klebel, Stefan Reichmann, Jessica Polka, Gary McDowell, Naomi Penfold, Samantha Hindle, Tony Ross-Hellauer.

**Methodology:** Thomas Klebel, Jessica Polka, Gary McDowell, Tony Ross-Hellauer.

**Project administration:** Tony Ross-Hellauer.

**Resources:** Jessica Polka.

**Supervision:** Tony Ross-Hellauer.

**Visualization:** Thomas Klebel.

**Writing – original draft:** Thomas Klebel, Stefan Reichmann, Jessica Polka, Gary McDowell, Tony Ross-Hellauer.

**Writing – review & editing:** Thomas Klebel, Stefan Reichmann, Jessica Polka, Gary McDowell, Naomi Penfold, Samantha Hindle, Tony Ross-Hellauer.

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
