## [Decision Letter · Decision Letter 0]

11 Jun 2020

PONE-D-20-10194

Peer review and preprint policies are unclear at most major journals

PLOS ONE

Dear Dr. Ross-Hellauer,

Thank you for submitting your manuscript to PLOS ONE. After careful consideration, we feel that it has merit but does not fully meet PLOS ONE’s publication criteria as it currently stands. Therefore, we invite you to submit a revised version of the manuscript that addresses the points raised during the review process.

First of all, apologies for the delay. I had trouble to allocate suitable (and available) referees for performing the review of your paper in the middle of the current worldwide crisis.

Your work has been assessed by two acknowledged experts in the field addressed by the paper. Both reviewers provided an overall positive feedback to your manuscript; however, there are some issues (most of them minor, and only a few relatively major) that you should respond through a Minor Revision of the paper. Concretely, reviewers (i) argue there are repetitive sentences along the text, some of them needing to be rephrased; (ii) point that some of the terms used in tables, cited resources and in-text paragraphs are not totally clear, and (iii) summarize an extensive set of specific comments that also should be attended by the authors.

We look forward to receiving your revised manuscript.

Kind regards,

Sergio A. Useche, Ph.D.

Academic Editor

PLOS ONE

Journal Requirements:

I have read the journal's policy and the authors of this manuscript have the following

competing interests:

Gary McDowell works at a for-profit that provides consulting services to organizations

addressing issues concerning early career researchers. Samantha Hindle is Content

Lead at bioRxiv, a preprint server for the biological sciences. Tony Ross-Hellauer is

Editor-in-Chief of the journal “Publications” (ISSN 2304-6775).

Reviewers' comments:

Reviewer's Responses to Questions

**Comments to the Author**

1. Is the manuscript technically sound, and do the data support the conclusions?

Reviewer #1: Yes

Reviewer #2: Yes

2. Has the statistical analysis been performed appropriately and rigorously? 

Reviewer #1: Yes

Reviewer #2: Yes

3. Have the authors made all data underlying the findings in their manuscript fully available?

Reviewer #1: Yes

Reviewer #2: Yes

4. Is the manuscript presented in an intelligible fashion and written in standard English?

Reviewer #1: Yes

Reviewer #2: Yes

5. Review Comments to the Author

Reviewer #1: I enthusiastically recommend publication of the manuscript, which quantifies a problem with the publishing industry that is both widespread and relatively easy to fix. I hope that it will be widely read and that journals that are currently exhibiting the problems it highlights will change their guidance accordingly.

The manuscript is well-written and understandable. The methods are detailed and quantitative, and the data are openly shared so that anyone can check the results. The conclusions are justified by the results, and the authors explicitly acknowledge the limitations of the study. In short, it is rigorous, readable, and important.

My one substantive concern is that the data shared on Zenodo are poorly organized and poorly annotated, which would likely make their reuse difficult. The HTML files are helpful in this regard and very user-friendly, but it is not clear to me, for example:

* Where I would find plain-language descriptions of the columns in the first two figures in 01-overview.html. I’m sure that the meaning of “bibjson.publisher” is clear to the authors, but it is not clear to me.

* Where I would find the raw data (preferably in csv format) for pretty much anything.

* How I could identify the individual points in part A of h-indices-1.png.

There is a ton of data here, presumably everything that was analyzed for this study, and I applaud the authors for sharing it. However, if I were looking for the data underlying a particular analysis, I wouldn’t know where to start. The examples above are just spot-checked; I have not attempted to be thorough, and I’m not suggesting just fixing these. What I am suggesting is overhauling the organization of the data so that an interested reader could easily find the raw data underlying any analysis along with a description of what they mean. I am, in other words, suggesting that the data be organized and structured in accordance with open data best practices, for example with the recommendations of Whitlock 2011 (TREE 26: 61-65 https://doi.org/10.1016/j.tree.2010.11.006) (this is just the relevant publication that I’m most familiar with; I’m sure there are more recent, and possibly more detailed, reviews).

Reviewer #2: The paper is sound technically but there is repetition. Also rather than present the data which would be new, editorial commentary is mixed in. I have been submitting papers for review for 50+ years. I am also the founding editor of a peer-reviewed journal. The paper has a tone of being authors who are junior who are frustrated with the review process. In many cases, peer review policies are not clear. But an author can contact the journal to clarify issues. It is becoming increasingly difficult to for editors to obtain reviewers. This paper is written from the author's perspective, but this perspective is only one that is important.

I take a dim view of co-review. When I sent a paper out for review, I expected the person to whom the review was sent to actually review the paper. Giving the paper to grad students for review was not desirable.

Here are specific comments:

lines 43-5. there is singular for "graduate student" followed by "their." This happens throughout the paper.

line 46. The pre-print issue is complex. It is possible that there can be no consistent policy. For example, publicizing the paper to journalists may be undesirable. But pre-prints in a professional working paper series may be permissible but this may depend on the working paper series.

line 60. I doubt the statement is valid.

line 149-150. In my discipline, there are a few top journals. Their policies are clear to the scholars in the discipline. Admittedly, junior scholars typically want to publish in these journals and are disappointed when their job market papers are not accepted by the top journals.

line 200. Would the editors not want to control co-review? Co-reviewers may not have the requisite knowledge to review the paper.

Table 1. I have trouble with some of the terms in the left column. Could a phrase be added with the term in italics?

Line 247. The issue of pre-print is becoming more complex with articles being posted in e.g. Web of Science in advance of publication. Would you consider these papers "published?"

6. PLOS authors have the option to publish the peer review history of their article (what does this mean?). If published, this will include your full peer review and any attached files.

Reviewer #1: No

Reviewer #2: No

---

## [Author Response · Author response to Decision Letter 0]

18 Aug 2020

Please see attached detailed document "Response to Reviewers"

---

## [Decision Letter · Decision Letter 1]

9 Sep 2020

Peer review and preprint policies are unclear at most major journals

PONE-D-20-10194R1

Dear Dr. Ross-Hellauer,

We’re pleased to inform you that your manuscript has been judged scientifically suitable for publication and will be formally accepted for publication once it meets all outstanding technical requirements.

Kind regards,

Sergio A. Useche, Ph.D.

Academic Editor

PLOS ONE

Additional Editor Comments (optional):

Reviewers' comments:

Reviewer's Responses to Questions

**Comments to the Author**

1. If the authors have adequately addressed your comments raised in a previous round of review and you feel that this manuscript is now acceptable for publication, you may indicate that here to bypass the “Comments to the Author” section, enter your conflict of interest statement in the “Confidential to Editor” section, and submit your "Accept" recommendation.

Reviewer #1: All comments have been addressed

Reviewer #2: All comments have been addressed

2. Is the manuscript technically sound, and do the data support the conclusions?

Reviewer #1: Yes

Reviewer #2: Yes

3. Has the statistical analysis been performed appropriately and rigorously? 

Reviewer #1: Yes

Reviewer #2: Yes

4. Have the authors made all data underlying the findings in their manuscript fully available?

Reviewer #1: Yes

Reviewer #2: Yes

5. Is the manuscript presented in an intelligible fashion and written in standard English?

Reviewer #1: Yes

Reviewer #2: No

6. Review Comments to the Author

Reviewer #1: The authors have addressed all reviewer comments to my satisfaction. I enthusiastically recommend publication.

Reviewer #2: I am in a fileld the paper indicates does not have good guidelines for publication. Yet it is my impression that the guidelines are clear to advanced graduate students and assistant professors. Publishing in the top journals is extremely important for promotion. It is clear that papers can be distributed as working papers. In the medical field, by contrast, it is as clear that prepublication is prohibited. Having said this, the paper's effort at quantification of the issues will be useful to some investigators. There are places where I have trouble with the grammar. Often, for example, "their" refers to a single individual.

7. PLOS authors have the option to publish the peer review history of their article (what does this mean?). If published, this will include your full peer review and any attached files.

Reviewer #1: No

Reviewer #2: No

---

## [Editor Report · Acceptance letter]

28 Sep 2020

PONE-D-20-10194R1 

Peer review and preprint policies are unclear at most major journals 

Dear Dr. Ross-Hellauer:

I'm pleased to inform you that your manuscript has been deemed suitable for publication in PLOS ONE. Congratulations! Your manuscript is now with our production department. 

Kind regards, 

on behalf of

Dr. Sergio A. Useche 

Academic Editor

PLOS ONE